# Green Extracts and UPLC-TQS-MS/MS Profiling of Flavonoids from Mexican Oregano (*Lippia graveolens*) Using Natural Deep Eutectic Solvents/Ultrasound-Assisted and Supercritical Fluids

**DOI:** 10.3390/plants12081692

**Published:** 2023-04-18

**Authors:** Manuel de Jesús Bernal-Millán, Miriam del Carmen Carrasco-Portugal, J. Basilio Heredia, Pedro de Jesús Bastidas-Bastidas, Erick Paul Gutiérrez-Grijalva, Josefina León-Félix, Miguel Ángel Angulo-Escalante

**Affiliations:** 1Centro de Investigación en Alimentación y Desarrollo A.C., Culiacán 80110, Mexico; mbernal221@estudiantes.ciad.mx (M.d.J.B.-M.);; 2Unidad de Investigación en Farmacología, Instituto Nacional de Enfermedades Respiratorias Ismael Cosío Villegas, Ciudad de México 14080, Mexico; 3Cátedras CONACYT-Centro de Investigación en Alimentación y Desarrollo A.C., Culiacán 80110, Mexico

**Keywords:** flavonoids, oregano, antioxidant capacity, extraction techniques, microcapsules, chromatographic profile

## Abstract

Mexican oregano (*Lippia graveolens*) is an important source of bioactive compounds, such as flavonoids. These have presented different therapeutic properties, including antioxidant and anti-inflammatory; however, their functionality is related to the quantity and type of compounds, and these characteristics depend on the extraction method used. This study aimed to compare different extraction procedures to identify and quantify flavonoids from oregano (*Lippia graveolens*). Emerging and conventional technologies include maceration with methanol and water, and ultrasound-assisted extraction (UAE) using deep eutectic solvents (DES) such as choline chloride-ethylene glycol, choline chloride-glycerol, and choline chloride-lactic acid. Supercritical fluid extraction using CO_2_ as a solvent was also studied. Six different extracts were obtained and the total reducing capacity, total flavonoid content, and antioxidant capacity by ABTS^•+^, DPPH^•^, FRAP, and ORAC were evaluated. In addition, flavonoids were identified and quantified by UPLC-TQS-MS/MS. Results showed that UAE-DES had the best extraction effect and antioxidant capacity using colorimetric methods. However, maceration-methanol was superior in compound content, and highlighting naringenin and phloridzin were the major compounds. In addition, this extract was microencapsulated by spray drying, which provided a protection feature of their antioxidant potential. Oregano extracts are rich in flavonoids and the microcapsules present promising results for future research.

## 1. Introduction

Mexican oregano (*Lippia graveolens*) is a plant that has taken importance in the scientific field for its high phytochemical content. Among these compounds, flavonoids stand out as hydrophilic compounds produced by plants to defend against biotic and abiotic factors [1]. Pharmacologically, these compounds are known for their low toxicity and antioxidant, antihypertensive, anti-inflammatory, antiproliferative, and antiviral properties [2]. Among the main flavonoids reported in oregano are quercetin, naringenin, and pinocembrin [3,4,5]. Their functionality is related to the compound type (structure) and concentration. However, these characteristics depend on the extraction method used; there are conventional and unconventional methods [6]. The traditional methods are simple to execute and lead to high extraction yields; however, they are not selective. The methodology developers have opted for using solvents such as ethanol, methanol, and acetone, and not just water to extract bioactive compounds. Parameters such as time, particle size, type of solvent, mass/volume relation, and temperature, among others, have been evaluated in conventional flavonoid extraction methods [7]. On the other hand, unconventional methods, also known as “green extraction methods” or emerging technologies, are new and promising techniques to overcome the limitations of classic extraction methods. Within these methods are ultrasound-assisted extraction using deep eutectic solvents (DES) and supercritical fluid extraction using CO_2_ as a solvent. The advantages they provide are reduced sample degradation and organic solvent consumption, contamination prevention, improved extraction efficiency, selectivity, and automation capability [8]. The extraction of bioactive compounds from rosemary, clove, green tea, and oregano by different environmentally friendly techniques has been reported. Nonetheless, little has been studied about the use of DES on polyphenols and even less on this genus of plants [9]. However, regardless of the extraction method, during the handling and extraction of these compounds, their stability, efficacy, and functionality against diseases caused by pathogens or free radicals may be compromised, affecting feasibility and profitability [10]. For this reason, microencapsulation is the technology used to protect bioactive compounds from degradation or oxidation, reduce incompatibility problems, and control the release of the encapsulated active compound [11]. This study aimed to compare different extraction procedures for identifying and quantifying oregano (*Lippia graveolens*) flavonoids using various conditions, including solvent extraction with conventional and emerging technologies. Additionally, the extracts were analyzed by the UPLC-TQS-MS/MS, and their antioxidant potential was measured using in vitro methods. Finally, the extract with the best chromatographic profile of flavonoids was microencapsulated and its antioxidant characterization was carried out.

## 2. Results and Discussion

### 2.1. Total Phenolic and Flavonoid Content

The total phenolic compounds content was higher in the ultrasound-assisted extraction using deep eutectic solvents (DES) (Table 1). The results ranged between 17.5 and 126.1 mg GAE/g, with the choline chloride-lactic acid extract showing the highest concentration. However, no significant differences were found between the choline chloride-glycerol and choline chloride-ethylene glycol extracts. The methanolic extract also obtained significant concentrations of total phenolics with 66.2 mg GAE/g, while the water and supercritical CO_2_ extracts had the lowest content with 27.9 and 17.5 mg GAE/g, respectively. The higher extraction capacity of flavonoids with DES can be attributed to the H-bond interactions between the phenolic compound molecules and the plant cell wall structure alteration by acoustic cavitation [12,13].

Regarding the total flavonoid content, a behavior similar to the phenolic compounds content was observed, with DES having the highest values. The total flavonoid content in order of extraction was as follows: ChCl-Et, ChCl-Gly, ChCl-LA, methanol, water, and supercritical CO_2_, with the choline chloride-ethylene glycol extract having the highest content (39.5 mg QE/g) and supercritical CO_2_ the one with the lowest content (5.2 mg QE/g). This behavior can be explained by the high polarity of DES and methanol, which is a less selective flavonoid extraction; in comparison, the supercritical CO_2_ technique is more selective, extracting greater amounts of specific flavonoids that are not easily detected with this colorimetric method [5].

### 2.2. Antioxidant Capacity by TEAC, FRAP, DPPH^•^, and ORAC Assays

The antioxidant capacity was determined by different methods, and the results are shown in Figure 1. The AC by ABTS^•+^ consists of the coloration intensity reduction of the ABTS^•+^ radical when reacting with an antioxidant compound. It ranged between 311.6 and 1090.9 μmol TE/g sample (Figure 1). The highest values were found in the ChCl-Gly, ChCl-LA, and ChCl-Et extracts (1090.9, 1074.4, and 1028.1 μmol TE/g), respectively, with no significant difference between them. The methanolic extract also showed high antioxidant potential with an antioxidant capacity of 801.6 μmol TE/g, followed by the water extract and supercritical CO_2_ (427.9 and 311.6 μmol TE/g), respectively, with significant differences between these extracts. The extract’s antioxidant capacity is associated with the composition and content of bioactive compounds in the extracts [14].

Regarding the results by FRAP, which consists of the reduction of the ferric ion, a similar trend was observed in antioxidant capacity by ABTS^•+^. The ChCl-LA extract (500.2 μmol TE/g) was the one that presented the highest antioxidant capacity, followed by the ChCl-Gly, ChCl-Et, methanol, water, and supercritical CO_2_ extracts (464.5, 407.2, 297.8, 118.2, and 56.3 μmol TE/g) respectively. The reducing power is related to the hydroxylation and conjugation degree of bioactive compounds. Therefore, DES’s high antioxidant capacity can be explained by their intermolecular interactions, mainly by hydrogen bonds between flavonoids and phenolic acids found in the solvent and the oregano extracts [15,16].

The DPPH^•^ assay was also performed, one of the most widely used assays in plant extracts, where an antioxidant with a weak A–H bond reacts with a free radical DPPH^•^ causing discoloration of the molecule [17]. This study showed the highest values in the ChCl-Gly, ChCl-Et, and ChCl-LA extracts (432.4, 391.3, and 327.4 μmol TE/g), respectively. These DES were reported by Alsaud et al. [18] as the solvents that provide the extract with higher antioxidant capacities. However, the methanolic extract also presented a high antioxidant capacity (311.3 μmol TE/g). On the other hand, the lowest results were observed in water (63.4 μmol TE/g) and supercritical CO_2_ (27.5 μmol TE/g) extracts. The low results in supercritical CO_2_ may be due to the dilution factor since the results are expressed in relation to the weight of the plant sample. For this technology, the sample weight is greater than the different methods.

The ORAC assay results, which measures the oxidative degradation of a fluorescent molecule induced by a generator of peroxyl radicals [19], were the following: ChCl-LA presented the highest antioxidant capacity (2090 μmol TE/g) while supercritical CO_2_ presented the lowest antioxidant capacity with 571.1 μmol TE/g. However, the methanolic extract showed a high antioxidant capacity (1899.9 μmol TE/g), with no significant difference with the ChCl-LA, ChCl-Gly, and ChCl-Et extracts. These results are very important since this method uses peroxyl radicals, the best model of antioxidant reactions with reactive oxygen species in vivo. This method provides continuous generation of radicals on a realistic time scale [20].

### 2.3. Identification and Quantification of Oregano (Lippia graveolens) Flavonoid Extracts by UPLC TQS-MSMS

Flavonoids were identified according to the fragments obtained in each sample spectrum with the spectra provided by standards. Sixteen flavonoid-type compounds were searched, of which fourteen were identified. In the methanolic extract, thirteen compounds were identified, eleven in the water extract, and nine in the supercritical CO_2_ extract, while in ChCl-Et, ChCl-Gly, and ChCl-LA extracts, eight, twelve, and nine compounds were detected, respectively. Three flavanones were found in our samples, namely naringenin, naringin, and hesperidin; the flavones vitexin, apigenin, luteolin, and luteolin 7-glucoside were also identified. Four flavonoids are flavonols: quercetin, quercitrin, rutin, and kaempferol; two dihydrochalcones identified as phloretin and phloridzin; and one isoflavone identified as genistein (Table 2). The presence of flavonoids in oregano, mostly flavones and flavanones such as apigenin, luteolin-7-glucoside, and naringenin, has been previously reported [21,22,23]. The quantification of flavonoids was carried out based on commercial standards. The compounds with the highest concentration for all types of extraction/solvent were quercetin, luteolin-7-glucoside, and hesperidin. Standing out with high concentrations were the flavanone, naringenin, and the dihydrochalcone, phloridzin. It is worth mentioning that naringenin has been previously described as predominant in *Lippia graveolens* extracts [5]. However, the flavonoid phloridzin has not been reported as a representative in oregano but in apple tree bark [24,25]. Anti-inflammatory, antioxidant, antidiabetic, and anticancer properties, among others, have been attributed to these compounds [26,27,28]. It is important to highlight that the methanolic extract was the one that obtained the best flavonoid profile, so it was decided to microencapsulate this extract.

### 2.4. Total Phenolic and Total Flavonoid Content in Oregano Microcapsules

The total phenolic and total flavonoid content in the oregano microcapsules were 17.3 mg GAE/g and 5.0 mg QE/g of sample, respectively, as compared to the non-microencapsulated extract, which showed 83.7 mg GAE/mg and 25.1 mg QE/mg dry extract content, respectively (Table 3). It should be noted that encapsulation efficiencies of 94.5% phenolic compounds and 90.1% total flavonoids were achieved. However, the phenolic content results in the microcapsules were higher than those reported by Bernal-Millan et al. [29]. They obtained 14.05 mg GAE/g of microcapsules, starting from oregano (*Lippia graveolens*) aqueous extract under the same operating conditions. Additionally, other studies, such as the one from Rezende et al. [30], reported the total phenolic compounds and flavonoids (10.16 mg GAE/g and 2.26 mg QE/g, respectively) content of *Malpighia emarginata* DC bioactive compounds microencapsulated by a spray dryer. These results were lower than those reported in our study. The values reported in the oregano microcapsules are lower than those found in the extract without microencapsulation. However, it has been reported that these differences are due to the dispersion of the spatial distribution of the polyphenols when adding an encapsulating polymer; therefore, the quantification per unit of mass decreases [31,32]. In addition, it has been shown that the microencapsulation of bioactive compounds, due to the decrease in water activity, can improve the target compounds with new physical characteristics or simply protect their physicochemical properties, ensure microbiological stability, and avoid degradation processes; it also reduces storage costs and allows a controlled release of the encapsulated active compound [29,33].

### 2.5. Antioxidant Capacity by ABTS^•+^, FRAP, and ORAC Assay in Oregano Microcapsules

Table 3 shows the oregano microcapsule’s antioxidant capacity values through the ABTS^•+^, FRAP, and ORAC tests. The results were 241.9, 178.6, and 637.0 μmol TE/g sample, respectively. These values were lower than those observed in the extract without microencapsulation for the three types of antioxidant activity assay, which is justified by the dilution of the extract when mixed with the wall material. In addition, the drying process did not affect the estimated antioxidant activity in the microcapsule, observing similar values to those expected.

Regarding the oregano bioactive compounds microencapsulated by spray drying, Bernal-Millan et al. [29] reported 50.83 μmol TE/g and 85.17 μmol TE/g for DPPH^•^ and ABTS^•+^ radicals’ inhibition. Our high values can be attributed to some modifications in the extraction methods, concentrating the extract, and therefore the microcapsules presented a greater antioxidant capacity. In other studies of the microencapsulation of bioactive compounds, Batista et al. [34] microencapsulated soy molasses isoflavones by spray drying using similar conditions to our research (Maltodextrin 18%, input temperature 140 °C), determining in the microcapsules an antioxidant capacity of 22.74 μmol TE/g through the ORAC assay, this value was much lower than that reported in oregano microcapsules. Additionally, Ydjeed et al. [35] encapsulated phenolic compounds from Carob (*Ceratonia siliqua* L.) pulp extracts, evaluating ORAC, FRAP, and DPPH^•^ capacity, reporting 295.8, 156.1, and 309.7 μmol TE/g, respectively, values below those obtained in the reported oregano microcapsules. These differences between the reported antioxidant capacity values can be attributed to the bioactive compound type and amount, source type (extract), process type, and encapsulating agent used.

### 2.6. Identification and Quantification of Flavonoids in Oregano (Lippia graveolens) Microcapsules

The chromatographic profile of the microencapsulated flavonoids and the extract without microencapsulating was determined. The results are shown in Table 4. Ten flavonoids were identified, four from the flavone type, two flavanones, two flavonols, one isoflavone, and one dihydrochalcone. However, there were three major compounds, in order of concentration, naringenin (1181.3 μg/g), phloridzin (677.9 μg/g), and quercetin (329.4 μg/g). These flavonoids have already been reported as the majority compounds except for phloridzin, which its presence has been associated with other matrixes such as apple trees [24]. Bernal-Millan et al. [29] identified 11 flavonoids in oregano (*Lippia graveolens*) microencapsulated phenolic extract and reported among its main compounds naringenin with 204.65 μg/g, reported that the value was five-times lower than that obtained in this study. However, these differences may be influenced by factors such as the sample nature and the fact that they did not use standards for their quantification analysis, reporting all its compounds in quercetin equivalents. Oregano microcapsules have great bioactive potential since they contain flavonoids with antioxidant, anti-inflammatory, diabetic, anticancer, and antihypertensive properties, among others.

Moreover, the bioactive benefits of polyphenols highly depend on their bioaccessibility and bioavailability, and the main reason for microencapsulation techniques is to increase the stability of polyphenols and enhance their bioaccessibility. Further studies should be focused on the release kinetics of polyphenols from Mexican oregano from these samples and their stability during the gastrointestinal tract.

### 2.7. Particle Morphology

Size and shape are the main characteristics to consider when producing microcapsules. The micrographs of the oregano microcapsules obtained by scanning electron microscopy showed a particle size between 2 and 10 μm and a spherical shape with flattening (Figure 2). In a study by Bernal-Millán et al. [29], oregano phenolic compounds extracted using water as a solvent were microencapsulated. They found similar features, the particles presented a spherical shape with depressions, and they mentioned that the roughness was attributed to the high drying speed, causing contractions in the microparticles due to the drastic moisture loss. They also obtained particles with sizes between 2 and 12 μm; these dimensions are typical of microcapsules obtained by spray drying. They stated that the degree of hydrolysis of maltodextrin and the pressure of the atomizing air influence the characteristics of the particles.

## 3. Materials and Methods

### 3.1. Chemicals and Raw Materials

The methanol was obtained from CTR Scientific. The carbon dioxide 99.9% (CO_2_) was bought in Lynde, Mexico. The choline chloride, glycerol, ethylene glycol, and lactic acid were obtained from Sigma Aldrich. The analytic standards quercetin (≥95%), vitexin (≥95%), apigenin (≥95%), quercitrin (88.4%), luteolin (≥98%), luteolin-7-glucoside (≥98%), naringenin (≥95%), naringin (≥95%), genistein (≥98%), rutin (94%), hesperidin (≥80%), kaempferol (≥97%), phloretin (≥99%), and phloridzin (≥99%) were from Sigma Aldrich (St. Louis, MO, USA). The mass spectrometry grade acetonitrile was acquired from JT Baker (Phillipsburg, NJ, USA). The oregano was obtained from the indigenous town Temohaya, at Mezquital, Durango, México (N: 23.299722; W: 104.509167). Oregano leaves were dried at 40 °C for 24 h in an Excalibur Food Dehydrator Parallax Hyperware (Sacramento, CA, USA) and minced in an IKA Werke M20 mill (Wilmington, NC, USA), obtaining a fine powder with a sieve #40. The oregano powder was stored at −20 °C until use.

### 3.2. Extraction of Flavonoids from Oregano (Lippia graveolens)

#### 3.2.1. Extraction of Flavonoids by Maceration

The rich flavonoid extracts were separated following the Chang et al. [36] methodology: 1 g of dried oregano and 10 mL of solvent (methanol and distilled water) were homogenized with an agitator SHAKER DOS-10L (Riga, Latvia) for 2 h in darkness. Subsequently, the extract was centrifuged at 10,000 rpm for 15 min at 4 °C; the supernatant was collected and stored at 4 °C for later use.

#### 3.2.2. Ultrasound-Assisted Extraction of Flavonoids

The extraction solvents were deep eutectic solvents (DES), synthesized according to Barbieri et al. [12], with some modifications. Choline chloride (ChCl) was used as hydrogen bond acceptor, and ethylene glycol (Et), glycerol (Gly), and lactic acid (LA) as hydrogen bond donors; the reagents were mixed and heated to 50 °C with 30% water. The synthesized DES were ChCl-Et, ChCl-Gly, and ChCl-LA at a 1:4 molar ratio. For the flavonoid extraction, 1 g of dried oregano and 10 mL of solvent (ChCl-Et, ChCl-Gly, ChCl-LA) were homogenized in an agitator. Subsequently, they were sonicated in a Branson ultrasonic CPX8800H-E (Brookfield, CT, USA) for 45 min at 45 °C and 40 kHz. Finally, the obtained extract was centrifuged at 10,000 rpm for 15 min at 4 °C; the supernatant was collected and stored at 4 °C for later use.

#### 3.2.3. Extraction of Flavonoids by Supercritical Fluid

The flavonoids were extracted following the protocol of Picos-Salas et al. [5]. An MV-10 ASFE (Waters Corp., Milford, MA, USA) extractor was used with CO_2_ as solvent and ethanol (99.9%) as a cosolvent in a 95:5 ratio. First, 2.5 g of sample were placed in a 10 mL beaker, and the following conditions were used: CO_2_ flow (4.75 mL/min), ethanol flow (0.25 mL/min), pressure at 245 bar, temperature at 36 °C, and static and dynamic extraction time (30 min, each). After the extraction, the extract was dried in a vacuum concentrator. Then, the yield was calculated, and the sample was resuspended in ethanol (99.9%) and stored at −20 °C for further analysis.

### 3.3. Antioxidant Analysis

#### 3.3.1. Trolox Equivalent Antioxidant Capacity (TEAC)

The TEAC assay determined the antioxidant capacity is based on absorbance inhibition of the ABTS^•+^ radical caused by the reaction with antioxidants, following the Thaipong et al. [37] methodology. The reaction solution was prepared by homogenizing 1 mL of 2.6 mM potassium persulfate and 1 mL of 7.4 mM ABTS^•+^. The mixture was left at room temperature for 16 h. For the assay, 10 μL of the extract was added to 190 μL of the reaction solution (ABTS^•+^). Subsequently, it was incubated in darkness for 2 h. Finally, the plate was read at 734 nm in a Synergy HT Microplate reader (BioTek, Inc., Winooski, VT, USA). Results were expressed in equivalent micromoles of Trolox per gram dry sample (μmol TE/g).

#### 3.3.2. Oxygen Radical Absorbance Capacity (ORAC)

For this test, the methodology of Huang et al. [38] was followed. Fluorescein was used as a fluorescent probe, Trolox as standard, and AAPH (2,2′-azobis dihydrochloride (2-amidino-propane)) as a peroxyl radical generator. The reaction mixture included 25 μL of extract, 75 μL of AAPH 95.8 μM, and 200 μL of 0.96 μM fluorescein in a 96-well microplate with a clear bottom and black walls; 75 mM phosphate buffer was used as blank. The fluorescein, samples, and phosphate buffer were preincubated at 37 °C for 15 min. Fluorescence loss was measured every 70 s for 70 min at a 485 nm wavelength for excitation and 580 nm for emission using a Synergy HT spectrophotometer. Values were calculated using a regression equation that describes the relationship between Trolox concentration and the net area under the fluorescein decomposition curve. To calculate the results, a Trolox curve from 6.25 to 125 (μmol TE/g) was used, and the results are expressed in equivalent μmol of Trolox per gram dry sample (μmol TE/g).

#### 3.3.3. Antioxidant Capacity by DPPH^•^ Assay

This method uses the radical 2,2-diphenyl-1-picrylhydrazyl (DPPH^•^) and reduces its purple chromogen by the action of an antioxidant compound to hydrazine, which colors a pale-yellow tone. The test was executed according to what was described by Karadag et al. [39]. First, in a 96-well plate, 10 μL of extract, 10 μL of Trolox standard curve, and 10 μL of blank (solvent) were added; then 190 μL of DPPH^•^ 200 μM reagent was added. It was incubated in darkness for 30 min. Finally, samples in the plate were read at 540 nm absorbance in a Synergy HT Microplate Reader (BioTek, Inc., Winooski, VT, USA). Results are expressed in equivalent μmol of Trolox per gram dry sample (μmol TE/g).

#### 3.3.4. Ferric Reducing Antioxidant Power Assay (FRAP)

This analysis is based on the power of a compound or an extract to reduce the ligand complex of ferric ions (Fe^3+^) to the deep blue-colored ferrous complex (Fe^2+^) in an acid medium. The FRAP analysis was performed as detailed by Ghasemzadeh et al. [40]. First, the FRAP reagent was prepared by mixing 1 mL of TPTZ 30 mM, 1 mL of FeCl_3_•6H_2_O 60 Mm, and 10 mL of acetate buffer. Then, in a 96-well plate, 30 μL of extract, 30 μL of Trolox standard curve, and 30 μL of blank (solvent) were added. 120 μL of FRAP reagent was added. It was incubated in darkness for 4 min. Finally, samples in the plate were read at 590 nm absorbance in a Synergy HT Microplate Reader (BioTek, Inc., Winooski, VT, USA). Results are expressed in equivalent μmol of Trolox per gram dry sample (μmol TE/g).

### 3.4. Total Phenolic and Flavonoid Content

#### 3.4.1. Total Phenolic Content

The total phenolic content, also known as total reducing capacity, was done to measure the total phenol content determined by the Folin–Ciocalteu reagent, following the methodology reported by Swain and Hillis et al. [41], with some modifications. First, the reaction mixture was prepared by mixing 10 μL of the sample, 230 μL of distilled water, 10 μL of Folin-Ciocalteu reagent, and 25 μL (2 M) of sodium carbonate solution. The reaction mixture was incubated for 2 h before absorbance was read at 725 nm using a Synergy HT 96-well microplate reader (Bio-Tek Instruments, Inc., Winooski, VT, USA). The results were expressed in equivalent milligrams of gallic acid per gram dry sample (mg GAE/g).

#### 3.4.2. Total Flavonoid Content

As reported by Chang et al. [36], the aluminum chloride colorimetric method determined total flavonoid content with slight modifications. First, 10 μL of the prepared extract, 10 μL of a blank (extraction solvent), and 10 μL of a quercetin standard curve were added to a 96-well plate. Subsequently, 250 μL of distilled water, 10 μL of 10% aluminum chloride, and 10 μL of 1 M potassium acetate were added. The sample was incubated in darkness for 30 min before reading the plate in a Synergy HT Microplate reader (BioTek, Inc., Winooski, VT, USA) at an absorbance of 415 nm. The results were reported in equivalent milligrams of quercetin per gram dry sample (mg QE/g).

### 3.5. Identification and Quantification of Flavonoids by UPLC-TQS-MS/MS

The identification and quantification of oregano flavonoids were carried out following the report by Bernal-Millan et al. [29] with some modifications, using an UPLC class H equipment (Waters Corp., Milford, MA, USA) coupled to a XEVO-TQS mass analyzer (Triple Quadrupole) using a BeH Phenyl 1.7 mm × 2.1 × 100 column at 40 °C. The compounds were separated with an elution gradient, solution A (water-ammonium formate 5 mM pH 3) and solution B (acetonitrile-0.5% formic acid) at a 0.3 mL/min flow rate. The procedure for gradient elution was as follows: 0 min, 90% (A); 5 min, 10% (A); 5.10 min, 90% (A); and 8 min, 90% (A). Electrospray ionization (ESI) was performed to analyze compounds; the parameters used were a 1.5 kV capillary voltage, a 30 V sampling cone, temperature at 500 °C, and desolvation gas 800 (L/h). The identification and quantification of flavonoids by UPLC were performed based on the retention time and wavelength peak area of maximum absorption. Multilevel calibration curves were performed using the standards apigenin, vitexin, quercetin, quercitrin, luteolin, luteolin-7-glucoside, naringin, phloretin, phloridzin, kaempferol, hesperidin, rutin, naringenin, and genistein (Appendix A). The flavonoid content was expressed in micrograms per gram dry sample.

### 3.6. Preparation of Oregano (Lippia graveolens) Microcapsules

Firstly, as previously reported, the extraction of oregano flavonoids was carried out by maceration using methanol as a solvent. Then, the extract was evaporated in a rotary evaporator; the dry extract was resuspended in 100 mL of distilled water. Subsequently, the feed solution was prepared. The 100 mL of extract (3.5 g of dry extract) were mixed with 16% maltodextrin 10 DE on a stirring plate (Thermo Scientific Cimarec, Waltham, MA, USA) until completely dissolved. The mixture was fed to a Yamato AD311S Spray Dryer with the following conditions: inlet temperature at 145 °C, feed flow of 5 mL/min, airflow at 0.32 m^3^/min, and an atomization pressure of 0.1 MPa. The microcapsules were recovered and stored for later analysis.

### 3.7. Flavonoid Encapsulation Efficiency and Particle Morphology

The encapsulation efficiency was calculated using the ratio between the theoretical flavonoid and the flavonoid content obtained in the microcapsules and reported as a percentage. Microparticle morphology was analyzed by an environmental scanning electron microscope model EVO-50 (Carl Zeiss, Oberkochen, Germany). The sample was placed using double-sided adhesive carbon tape on a sample holder without previous treatment. Observation was made with a secondary electron detector (SE1) and a 10–15 kV acceleration voltage (×2000 and ×4000 magnification) under high vacuum conditions.

### 3.8. Statistical Analysis

All experiments had three replicates. Data were expressed as the mean ± standard deviation (SD). Significant differences in evaluated parameters among different samples were analyzed by Minitab software (version 21.0, Minitab LLC, State College, PA, USA). The probability value of *p* < 0.05 was considered significant.

## 4. Conclusions

It was found that both conventional methods and emerging extraction technologies of bioactive compounds such as flavonoids present high extraction yields; however, the methanol-maceration method and the ultrasound-assisted extraction-DES showed the best antioxidant results. Microencapsulation of oregano flavonoids was also carried out, achieving a 90.1% encapsulation; various tests determined that the microcapsules presented antioxidant potential. Additionally, 14 flavonoids were identified and quantified, highlighting naringenin and phloridzin, reported with high bioactive potential. For all the above, the different oregano (*Lippia graveolens*) extracts, which are rich in flavonoids, and the microcapsules present promising results to investigate their potential against various diseases.

## Figures and Tables

**Figure 1 plants-12-01692-f001:**
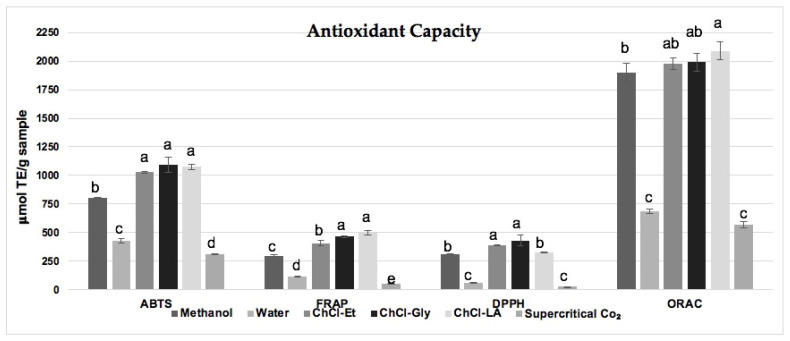
Antioxidant capacity of different oregano extracts (*Lippia graveolens*) by ABTS^•+^, FRAP, DPPH^•^, and ORAC assays. Distinct letters in the same assay bars show the significant difference (*p* < 0.05).

**Figure 2 plants-12-01692-f002:**
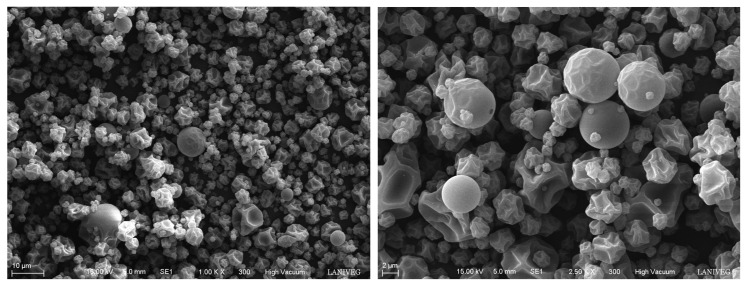
Micrographs of the microparticle structure of oregano (*Lippia graveolens*) flavonoids produced with maltodextrin 10 DE as wall material, using spray drying.

**Table 1 plants-12-01692-t001:** Total phenolic and total flavonoid contents in oregano extracts (*Lippia graveolens*).

Sample/Extracts	Total Phenolic (mg GAE/g)	Total Flavonoids (mg QE/g)
Methanol	66.2 ± 3.4 ^b^	26.9 ± 1.8 ^bc^
Water	27.9 ± 3.4 ^c^	19.2 ± 0.3 ^c^
ChCl-Et	100.1 ± 7.5 ^ab^	39.5 ± 3.3 ^a^
ChCl-Gly	123.6 ± 9.8 ^a^	39.4 ± 2.9 ^a^
ChCl-LA	126.1 ± 6.0 ^a^	37.3 ± 3.0 ^ab^
Supercritical CO_2_	17.5 ± 1.3 ^c^	5.2 ± 0.8 ^d^

GAE: gallic acid equivalents; QE: quercetin equivalents; ChCl: choline chloride, Et: ethylene glycol; Gly: glycerol; LA: lactic acid. Data are shown as means ± standard deviation of three replicates (*n* = 3). Different superscript letters in the same column indicate a significant difference (*p* < 0.05).

**Table 2 plants-12-01692-t002:** Identification and quantification of flavonoids in oregano (*Lippia graveolens*) extracts.

Compounds	Compound Type	Method/Solvent					
		Maceration/ Methanol	Maceration/ Water	Ultrasound-Assisted/ ChCl-Et	Ultrasound-Assisted/ ChCl-Gly	Ultrasound-Assisted/ ChCl-LA	Supercritical/ CO_2_-Ethanol
Quercetin	Flavonol	290.04 ± 2.88 ^b^	8.92 ± 0.91 ^c^	3.54 ± 0.85 ^d^	9.93 ± 3.16 ^c^	2.60 ± 1.61 ^d^	432.69 ± 0.14 ^a^
Vitexin	Flavone	ND ^b^	ND ^b^	ND ^b^	1.24 ± 0.63 ^a^	1.30 ± 0.09 ^a^	ND
Apigenin	Flavone	15.58 ± 3.05 ^b^	1.03 ± 0.25 ^c^	ND ^c^	ND ^c^	ND ^c^	22.22 ± 3.31 ^a^
Quercitrin	Flavonol	12.57 ± 2.03 ^ab^	4.59 ± 0.70 ^c^	13.86 ± 1.45 ^a^	10.05 ± 1.33 ^b^	4.66 ± 0.88 ^c^	ND ^d^
Luteolin	Flavone	181.28 ± 36.69	26.79 ± 3.73	ND	3.58 ± 0.80	ND	7.63 ± 1.06
L7G	Flavone	1247.84 ± 68.73 ^b^	489.54 ± 15.09 ^e^	1334.14 ± 6.78 ^a^	664.22 ± 21.29 ^d^	765.73 ± 17.83 ^c^	1.15 ± 0.26 ^f^
Naringenin	Flavanone	3505.74 ± 163.60 ^a^	452.48 ± 22.21 ^b^	430.31 ± 3.46 ^b^	286.85 ± 72.07 ^b^	102.33 ± 21.13 ^c^	3513.78 ± 113.69 ^a^
Naringin	Flavanone	1.00 ± 0.10 ^b^	ND ^b^	1.50 ± 0.36 ^b^	1.49 ± 0.24 ^b^	4.77 ± 1.42 ^a^	ND ^b^
Genistein	Isoflavone	15.44 ± 1.91 ^b^	ND ^c^	ND ^c^	ND ^c^	ND ^c^	24.03 ± 3.44 ^a^
Rutin	Flavonol	2.37 ± 0.92 ^de^	5.25 ± 0.09 ^d^	36.95 ± 2.61 ^a^	30.01 ± 1.19 ^b^	11.24 ± 3.48 ^c^	ND ^e^
Hesperidin	Flavanone	25.56 ± 3.58 ^b^	11.70 ± 2.64 ^bc^	101.65 ± 2.94 ^a^	85.81 ± 2.65 ^a^	81.56 ± 31.56 ^a^	ND ^c^
Kaempferol	Flavonol	12.11 ± 2.40 ^ab^	9.83 ± 0.27 ^b^	ND ^d^	2.56 ± 0.04 ^c^	ND ^d^	14.22 ± 1.5 ^a^
Phloretin	Dihydrochalcone	6.26 ± 0.17 ^b^	7.50 ± 0.23 ^b^	ND ^c^	1.61 ± 0.32 ^c^	ND ^c^	37.78 ± 2.46 ^a^
Phloridzin	Dihydrochalcone	4068.91 ± 50.59 ^a^	1306.56 ± 49.05 ^d^	3049.26 ± 80.93 ^b^	1773.93 ± 23 ^c^	1177.2 ± 100.32 ^e^	73.61 ± 8.26 ^f^

ChCl: choline chloride; Et: ethylene glycol; Gly: glycerol; LA: lactic acid; ND: not detected. Results are expressed in micrograms per gram of dry sample (μg/g). Results are shown as means ± standard deviation of three replicates (*n* = 3). Different superscript letters in the same row indicate the significant difference (*p* < 0.05).

**Table 3 plants-12-01692-t003:** Total phenolic compounds, flavonoids, and antioxidant capacity ABTS^•+^, FRAP, and ORAC from oregano (*Lippia graveolens*) extract and microcapsules.

Sample	TPC (mg GAE/g Sample)	TFC (mg QE/g Sample)	ABTS^•+^ (μmol TE/g Sample)	FRAP (μmol TE/g Sample)	ORAC (μmol TE/g Sample)
Microcapsules	17.3 ± 2.1	5.0 ± 0.3	241.9 ± 4.7	178.6 ± 2.6	637.0 ± 4.6
Non-microencapsulated extract	83.7 ± 5.9	25.1 ± 2.9	1198.6 ± 55.5	992.6 ± 20.8	4153.8 ± 133.1

TPC: total phenolic content; TFC: total flavonoid content; GAE: gallic acid equivalents; QE: quercetin equivalents; TE: Trolox equivalents. Data shown as means ± standard deviation of three replicates (*n* = 3).

**Table 4 plants-12-01692-t004:** Identification and quantification of flavonoids in oregano (*Lippia graveolens*) microcapsules.

Compounds	Compound Type	Microcapsules (μg/g)	Non-Microencapsulated Extract (μg/g)
Quercetin	Flavonol	329.43 ± 28.73	1916.0 ± 164.61
Vitexin	Flavone	ND	ND
Apigenin	Flavone	22.40 ± 2.21	53.23 ± 7.34
Quercitrin	Flavonol	ND	ND
Luteolin	Flavone	17.29 ± 0.58	64.24 ± 1.18
Luteolin-7-glucoside	Flavone	67.48 ± 1.74	111.63 ± 8.47
Naringenin	Flavanone	1181.35 ± 104.23	3840.11 ± 199.92
Naringin	Flavanone	ND	ND
Genistein	Isoflavone	15.00 ± 2.70	80.25 ± 0.97
Rutin	Flavonol	3.04 ± 0.15	8.54 ± 0.73
Hesperidin	Flavanone	3.49 ± 0.58	10.09 ± 0.46
Kaempferol	Flavonol	17.33 ± 1.66	82.11 ± 4.54
Phloretin	Dihydrochalcone	ND	ND
Phloridzin	Dihydrochalcone	677.92 ± 6.72	1202.08 ± 35.44

ND: not detected. Results are shown as means ± standard deviation of three replicates (*n* = 3).

## Data Availability

All data are included in the main text.

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
