# Peer review of "Green Extracts and UPLC-TQS-MS/MS Profiling of Flavonoids from Mexican Oregano (Lippia graveolens) Using Natural Deep Eutectic Solvents/Ultrasound-Assisted and Supercritical Fluids"

_plants, 2023, doi:10.3390/plants12081692_

Round 1

Reviewer 1 Report

Here are my suggestions for authors, separated as general and specific comments (given with an appropriate Line number(s) from text in order to facilitate tracking).

General comments

1. In science there are no "we", "our", etc. So, use of personal pronouns should be avoid as much as is possible. In that sense, I would like to ask authors to revise/check a whole document.

2. Some parts of text are hard to follow. Suggest to authors to check English during revision.

3. ABTS is a radical cation while DPPH is a radical and not completely same. Both must be labeled uniformly and in the same way through a whole text. Please carefully revised document.

4. Introduction section is lack in data about previous similar research on oregano or similar species. Improve it.

5. Apart from type of extraction and similar on flavonoids bioactivity an important parameter that can influence is their bioaccessibility/bioavailability which can be quite limited in our body. This topic is one of the most popular in current research. This important issue is completely overlooked in research. Please perform literature search and include additional discussion whenever is needed in text.

Specific comments

Line 14: Provide Latin plant name here. This is the first mention in text. In addition, I think that authors should specify that this is not Origanum vulgare which is know wide world as oregano but this is  so-called "orégano chiquito" or "Mexican oregano (https://doi.org/10.3390/molecules26175156)?

Lines 18-21: Please check English and grammar. Sentence is too long and hard to understand.

Line 26: Is this "m-methanol" a typo? Check/correct.

Line 26: This phloridzin is litlle bit confusing to me since it is known that it is chemotaxonomic marker for Malus genus or for some plants from Rosaceae family (https://doi.org/10.1016/j.phytochem.2010.03.003). But your plant does not belongs to neither this genus nor family. So, can authors find/provide literature data about similar results for plants from this genus/family where your oregano belongs?

Line 34: Latin name here also.

Line 52: "find" where, what? It is completely unclear what you are trying to say here. Please clarify/correct.

Lines 57-59: As I explain in general comments I think that some comments about bioaccessibility/bioavailability of these flavonoids must be included in your text.

Line 63: Put term "in vitro" in Italic.

Line 67: Correct to be "Total phenolic and flavonoid content".

Line 73: Suggest to replace "good" with "significant" here. It seems to me much more appropriate in this context.

Line 82: "QE" not "EQ". Correct.

Lines 83-85: I completely disagree with this explanation. MeOH, both pure or in some aqueous solutions, is an excellent solvent for all phenolics, including flavonoids. In addition, your results in HPLC section completely disagree with this your statement, right? Here is more "problem" in spectrophotometry and its possible imprecision since it can not always "target" only one specific compound(s). In aprticular Folin assay is well known as non-specific since it determines not only phenolic but all redox substances including bioactive peptides, reducing sugars, some micro elements, etc.

Lines 92-100: This is actually TEAC assay and it should be labeled in that way, in order to avoid confusion among readers. SO, please do not use "AC" abbreviation.

Lines 100-101: This is not totally true. Phenolics are not the only one antioxidants in plant material. There are also, carotenoids, peptides, some sugars and some other bioactive compounds. Correct/expand this statement.

Line 111: If DPPH radical is "stable" that it would not react with anything, rignt? DPPH is not stable but during reaction with antioxidants it creates stable product. Please correct.

Line 128: Put term "in vivo" in Italic here.

Line 134: Latin name before not after word "flavonoid".

Lines 142-143: Rutin and quercetin can not belong to different sub-groups since rutin is quercetin derivative. Both are flavone not flavonol for quercetin. Correct.

Line 151: Latin name in Italic here.

Lines 151-153: Again, same comment for phoridzin.

Line 159: Delete Latin name. It is completely surplus here.

Lines 160-163: I do not understand how 17.3 mg/g is 94% of 83.7 mg/g?? Please clarify/expand/correct.

Line 165: typo- missing "mg" in front of "GAE" here? Check/correct.

Line 168: typo - "flavonoids" in plural here. In addition, put Latin name in Italic here.

Line 173: "phenolic" not "polyphenolic". It is "polyphenols" in that form. Correct.

Line 177: "physico-chemical". Correct.

Line 187: "TE" not "ET". The error is repeating in several next Lines. Please correct all.

Line 192: What is the meaning of this "to those expected"? Explain/elaborate.

Line 202: Latin name in Italic here.

Lines 216 and 224: The same as previous.

Lines 225-226: I do no understand is this MeOH enxtract non-encapsulated, presented in the Table 4, the same as from Table 2 or not? If it is than why you give again results which are now different compared to Table 2? If it is different, why you made again the same MeOH extract and how it can have different results for HPLC?

Line 235: "stated" in past tense here.

Line 255: Put Latin name in brackets here.

Lines 283-298: The given assays are not antioxidant assays but phytochemical assays. Please exclude from subsection 3.3.

Line 288: Please do not use normality as concentration unit. It is archaic and not acceptable in SI system. Replace with molarity (M).

Lines 283-357: For all assays as well as for HPLC analysis authors should specify did they express obtained results based on fresh or dry weight? It is quite important to know.

Kind regards.

Author Response

Please, attached find the response letter. Thank you.

Reviewer 2 Report

The article is very interesting and gives an insight into the bioactive compounds and antioxidant activity of oregano. In addition, it optimizes extraction methods, indicating those methods that extract the highest amount of phenolic compounds.

However, some formatting should be corrected. The tables should be revised so that they are complete on one page, as it would make reading easier.

In figure 1 the caption should be separated from the legend

Author Response

(The authors gave the same response as above.)

Reviewer 3 Report

The subject of the research presented in the manuscript is interesting. However, the study's experimental and data analysis parts are imprecise, for example, not a single chromatogram is presented throughout the whole manuscript. Therefore the readers don't have a chance to compare the phytochemical profiles (in terms of quality and quantity) of obtained extracts.

Author Response

(The authors gave the same response as above.)

Round 2

Reviewer 1 Report

Manuscript is improved. Some technical issues still should be corrected:

1. You have now labeled ABTS as cation which is not true. As I said ABTS is both radical and cation and should be labeled in tat way. Please revise Manuscript again and correct all technical issues.

2. DPPH should be always labeled as radical. Currently it is not. Correct.

3. In the first line of Introduction it is still just "Oregano" and it should be "mexican oregano" to avoid any confusion.

Lines 57-58: Please provide reference(s) for added statement.

Line 251: typo - "... release kinetic of polyphenols from Mexican oregano..." Correct. Your are not releasing oregano per se as it is stated currently.

Please correct.

Kind regards.

Author Response

Attached please find the response letter (second round).
